# Non-Destructive Identification of Dyes on Fabric Using Near-Infrared Raman Spectroscopy

**DOI:** 10.3390/molecules28237864

**Published:** 2023-11-30

**Authors:** Mackenzi Peterson, Dmitry Kurouski

**Affiliations:** 1Department of Entomology, Texas A&M University, College Station, TX 77843, USA; bigmack2002@tamu.edu; 2Department of Biochemistry and Biophysics, Texas A&M University, College Station, TX 77843, USA

**Keywords:** near-infrared Raman spectroscopy, fabric, colorants, chemometrics

## Abstract

Fabric is a commonly found piece of physical evidence at most crime scenes. Forensic analysis of fabric is typically performed via microscopic examination. This subjective approach is primarily based on pattern recognition and, therefore, is often inconclusive. Most of the fabric material found at crime scenes is colored. One may expect that a confirmatory identification of dyes can be used to enhance the reliability of the forensic analysis of fabric. In this study, we investigated the potential of near-infrared Raman spectroscopy (NIRS) in the confirmatory, non-invasive, and non-destructive identification of 15 different dyes on cotton. We found that NIRS was able to resolve the vibrational fingerprints of all 15 colorants. Using partial-squared discriminant analysis (PLS-DA), we showed that NIRS enabled ~100% accurate identification of dyes based on their vibrational signatures. These findings open a new avenue for the robust and reliable forensic analysis of dyes on fabric directly at crime scenes. Main conclusion: a hand-held Raman spectrometer and partial least square discriminant analysis (PLS-DA) approaches enable highly accurate identification of dyes on fabric.

## 1. Introduction

Forensic examination of a crime scene includes a detailed analysis of all pieces of evidence. Among all physical evidence, fiber can be used to link suspects to crime scenes or demonstrate the absence of such connections. In recently reported criminal cases, fiber evidence found on the victim’s body was analyzed [1,2]. The analysis was able to determine the manufacturers of the fabric. This information helped the investigators to make a connection between the suspect and the crime scene.

Current methods of fabric analysis primarily rely on pattern recognition and are performed by an expert using a microscope [3,4,5,6]. This subjective approach can be used to reveal fiber length, diameter, cross-sectional shape, and color, as well as the surface contour. Although the results of such subjective fiber analyses can be brought in a court of law, the U.S. Department of Justice reports that the production of synthetic textiles has made it more difficult to unambiguously identify the origin of a fiber [7]. Therefore, the National Academy of Sciences Report on Forensic Science states that more robust and reliable methods are needed to identify the origins of fibers and minimize both human error and sample contamination [8].

Several research groups demonstrated that Raman spectroscopy (RS) could be used for the identification of fiber material and fiber dyes [9,10,11]. RS is based on the phenomenon of inelastic scattering of light [12]. Inelastic scattering occurs because of energy exchange between the incident photons and the molecules present in the sample. Furthermore, the exchange in the photon directly depends on the chemical structure of the molecules [13]. Consequently, an acquisition of the inelastically scattered photons can be used to identify the molecular structure and composition of specimens. RS has previously been used to identify urine, sweat, semen, blood, hair colorants, and gunshot residue [14,15,16,17,18,19]. It should be noted that RS is a non-destructive and non-invasive technique [20,21,22]. Therefore, it can be used to identify the origin of highly vulnerable historical fabrics that have dyes or pigments [9]. In this case, the identification of colorants can help conservationists understand which substances can be used to restore artwork [23,24].

Experimental findings reported by Casadio and co-workers demonstrated that colored fabric often exhibits strong fluorescence, which obscures the possibility of using RS for the identification of both fabric material and dyes [11]. Van Duyne’s group demonstrated that noble metal nanostructures could be used to both suppress the fluorescence and enhance the Raman scattering from dye molecules present in fabric [25,26,27]. This analytical approach is known as surface-enhanced Raman spectroscopy (SERS) [28,29,30,31,32,33]. Kurouski’s group previously used SERS to identify colorants on hair that could not be directly detected using RS [34,35,36,37,38]. Furthermore, SERS is broadly used to detect explosives [39,40], illicit drugs such as heroin and cocaine [41], as well as body fluids [42]. Our group showed that SERS could be used to detect and identify artificial colorants on hair [34,43]. Furthermore, SERS could be used to track the degradation of colorants on hair triggered by UV, heat, and hypolimnion water [35,36,37].

We hypothesized that the problem of dye fluorescence can be overcome using near-infrared excitation in RS. Previously reported results in our group demonstrated that the use of 830 nm electromagnetic radiation could be used to overcome the problem of chlorophyll fluorescence in the Raman-based analysis of plant leaves [44]. Expanding upon this, we determine the extent to which near-infrared RS (NIRS) could be used to detect and identify colorants on fabric. Our findings demonstrate that a hand-held NIR spectrometer can reveal vibrational signatures of colored cotton with 15 different dyes. Furthermore, using chemometric approaches, these vibrational signatures allow for ~100% accurate detection and identification of the artificial colorants on fabric. 

It should be noted that experimental results obtained using NIRS can be testified in a court of law because this method meets all the requirements for the Daubert Standard, a list of requirements any method must undergo to become admissible in a court of law. The three requirements of Daubert are (1) the testimony is based on sufficient facts or data, (2) the testimony is a product of reliable principles and methods, and (3) the witness has applied the principle and methods reliably to the facts [45]. Some of the uses of Raman in court cases can be found in the cases of Takeda Pharmaceutical Company Ltd. (Tokyo, Japan) v. Teva Pharm. (Parsippany-Troy Hills Twp, NJ, USA) and Warner Chilcott Labs. Ireland Ltd. (Dundalk, Ireland) v. Impax Labs., Inc. (Hayward, CA, USA) [46,47].

## 2. Results and Discussion

### 2.1. NIRS-Based Analysis of Colored Fabric

NIRS spectra acquired from in-lab-colored and uncolored (blank) fabric exhibited strong vibrational bands; Figure 1 and Appendix A, and Table 1. The spectrum of uncolored fabric has vibrational bands at 379, 437, 494, 518, 566, 610, 723, 900, 968, 995, 1095, 1118, 1149, 1237, 1289, 1336, 1378, 1475, and 1602 cm^−1^, which primarily originate from cellulose [19,48]. Raman spectra acquired from the red-dyed cotton exhibited distinctly different vibrational fingerprints with bands at 648, 828, 1227, 1278, 1331, 1353, 1413, 1469, and 1611 cm^−1^. The orange-dyed cotton showed a spectrum with bands at 652, 829, 1256, 1415, 1565, and 1611 cm^−1^, whereas the yellow-dyed cotton exhibited a uniquely vibrational band centered at 1409 cm^−1^. We observed vibrational bands at 749, 815, 1181, 1265, 1407, and 1541 cm^−1^ in the spectra acquired from the green-dyed cotton, whereas the blue-dyed cotton had a unique set of vibrational bands at 602, 749, 1181, 1265, 1407, and 1541 cm^−1^. Finally, purple-dyed cotton exhibited vibrational bands at 701, 764, 833, 1033, 1243, 1407, 1500, and 1590 cm^−1^. Based on these results, we can conclude that fabrics colored with different dyes exhibit drastically different vibrational fingerprints. Thus, with this spectroscopic library, NIRS can be used to detect and identify a large variety of dyes on fabric.

We also found that in addition to the unique vibrational bands that were evident for all colored fabric, we found drastic differences in the intensity of most of the vibrational bands that were present in all acquired spectra. The most notable changes in intensities were found in the region of 1000–1200 cm^−1^. Thus, not only a unique vibrational fingerprint but also a change in the intensity of vibrational bands can be used to identify the colorants on fabric; Appendix A.

Next, we used partial least square discriminant analysis (PLS-DA) to determine the extent to which the above-discussed differences in the NIRS spectra could be used to identify fabric colored with different dyes. The LV plot can be found in Appendix A. We found that the PLS-DA model was able to predict all samples with 100% accuracy, except yellow- and plain-colored fabric, which were predicted with 96% and 84% accuracy, respectively; Table 2. High similarities between the Raman spectra acquired from yellow- and plain-colored fabric confirm this observation; Figure 1. To overcome this issue, we developed a separate PLS-DA model that was able to differentiate these two types of colored fabric with 100% accuracy; Table 3 and Appendix A. These results demonstrate that RS coupled to PLS-DA can be used for the highly accurate identification of colorants on fabric.

### 2.2. Spectroscopic Analysis of Dyes

We acquired NIRS spectra from the solution of dyes used to color the fabric to demonstrate that the discussed above vibrational bands originated from these molecules. To increase the signal-to-noise ratio, we used the spatial-offset (SORS) modality of the hand-held NIR Raman spectrometer. SORS allows a larger volume of the liquid to be probed compared to normal RS [49,50]. The red colorant exhibited a SORS spectrum with vibrational bands at 447, 536, 649, 830, 908, 981, 1156, 1328, 1354, 1417, 1469, and 1573 cm^−1^ (±5 cm^−1^), whereas the spectrum of the orange dye had bands at 379, 437, 454, 519, 567, 611, 654, 722, 828, 901, 974, 995, 1096, 1119, 1150, 1236, 1255, 1292, 1336, 1378, 1414, 1470, 1565, and 1608 cm^−1^; Figure 1, Appendix A, and Table 4. The SORS spectrum of the yellow dye contained peaks at 447, 616, 980, and 1609 cm^−1^, whereas the spectrum of the green dye exhibited vibrational bands at 448, 723, 749, 981, 1158, 1183, 1265, 1340, 1448, 1543, and 1607 cm^−1^. The blue dye produced new peaks at 447, 501, 605, 653, 727, 750, 981, 1097, 1157, 1186, 1266, 1339, 1448, 1542, and 1604 cm^−1^. Finally, the purple liquid dye produced new peaks at 406, 444, 633, 702, 762, 833, 907, 981, 1011, 1132, 1181, 12,009, 1244, 1500, and 1589 cm^−1^; Figure 1, Appendix A, and Table 4. It should be noted that different dyes not only exhibited unique vibrational signatures in SORS spectra but also had different intensities of the vibrational bands that were present in several spectra; Appendix A. These results also revealed the direct relationship between the vibrational bands observed in the NIRS spectra of colored fabric and the dyes themselves.

PLS-DA was used to determine the accuracy of dye identification; Table 5 and Appendix A. We found that the PLS-DA model was able to predict every colorant with 100% accuracy. These results demonstrated that dyes could be identified using machine learning via their unique vibrational fingerprints.

### 2.3. Raman-Based Analysis of Colored Menil Fabric

To further explore the potential of NIRS, we requested a colored Menil fabric from The Museum of Fine Arts Houston. Using a hand-held Raman spectrometer, we acquired Raman spectra from different dyes on the fabric, as well as the uncolored fabric itself. The Raman spectrum collected from uncolored fabric exhibited vibrational bands at 378, 439, 518, 566, 609, 722, 901, 969, 994, 1097, 1116, 1238, 1288, 1336, 1378, 1413, 1474, and 1784 cm^−1^; Figure 2 and Appendix A and Table 6. We also found that the Raman spectrum collected from fabric colored with Brazilwood had vibrations at 711, 848, 1030, 1150, 1206, 1512, and 1565 cm^−1^. The spectrum acquired from the fabric colored with Buckthorn were at 651, 739, 808, 853, 1026, 1202, 1552, 1603, and 1716 cm^−1^, whereas that acquired from the fabric colored with Clutch were at 677, 711, 748, 816, 858, 1026, 1150, 1204, 1517, 1556, 1603, 1721, 1820, 1851, 1877, 1909, 1942, and 1965 cm^−1^. We also observed vibrational bands at 663, 760, 1033, 1151, 1300, 1327, 1714, and 1913 cm^−1^ in the Raman spectrum collected from the fabric colored with Cochineal dye. Lac-colored fabric exhibited peaks at 504, 576, 616, 654, 803, 848, and 1601 cm^−1^ in the collected Raman spectrum, whereas Logwood-dyed material had peaks appear at 644, 736, 777, 804, 851, 1033, 1151, 1200, 1481, 1564, 1613, 1714, 1880, 1915, and 1964 cm^−1^ in the acquired Raman spectrum. We also observed a unique vibrational fingerprint for Madder dye on cotton at 616, 654, 803, 848, and 1601 cm^−1^ and Pomegranate at 751, 806, 1204, 1587, and 1716 cm^−1^. Finally, Weld-colored fabric exhibited a spectrum with vibrational bands at 650, 743, 798, 1205, and 1580 cm^−1^; Figure 2 and Appendix A. We also observed drastic differences in the intensities of the above-discussed vibrational bands. The major areas of intensity variation were in the regions of 350–500 cm^−1^, 1000–1400 cm^−1,^ and 1500–1650 cm^−1^; Appendix A. These results confirm that NIRS is highly sensitive to the differences in the chemical structure of different colorants. Therefore, we can conclude that NIRS can be used to identify various colorants on fabric.

PLS-DA was then used to analyze the acquired NIRS spectra. We found that PLS-DA was able to identify colorants present on the fabric with ~95% accuracy on average; Table 7. Specifically, the spectra acquired from the uncolored fabric were correctly identified with 92% accuracy, whereas Brazilwood and Buckthorn were predicted with 72% and 84% accuracies, respectively. Madder colorant was identified on fabric with 84% accuracy, whereas both Pomegranate- and Weld-colored fabric were identified with 96% accuracy. All other colorants were predicted with 100% accuracy. These results demonstrated that the coupling of RS with PLS-DA can be used for the highly accurate identification of different colorants on fabric. The LV plots of this model are found in Appendix A.

### 2.4. Discussion

The confirmatory identification of colorants on forensic-related pieces of evidence is a challenging task. The use of Raman spectroscopy is often limited to a very low amount of the colorants present in the sample and/or by a high sample fluorescence. The former issue can be overcome using AuNPs to both enhance Raman scattering and suppress the dye’s fluorescence. Our current results show that the high fluorescence of the colorants can be overcome using near-infrared electromagnetic radiation in Raman measurements. Specifically, we found that 830 nm excitation can be used to acquire excellent signal-to-noise Raman spectra from cotton fabric colored with 15 different dyes. We also showed that colorants exhibited unique vibrational fingerprints that can be used to identify the dyes. These unique spectroscopic signatures can be used to build a library that, together with chemometric algorithms, will enable the confirmatory identification of colorants on fabric.

It is important to emphasize that additional studies are required to understand the role of the fabric material in the NIRS-based identification of colorants. Elucidation of the role of other factors, such as fabric contamination with substances of biotic and abiotic origin, is also required to develop the use of NIRS for forensic and art conservation applications [51,52,53]. This work is currently in progress in our group.

## 3. Materials and Methods

Fabric materials: For the in-lab-colored fabric, 100% cotton canvas acquired from Fruit of the Loom was used. The pigment for the dyes was bought from the Mosaiz store. The pigments were received in powder form and were developed into colorants following manufacturing instructions. Specifically, the dyes were mixed with water and vigorously stirred until a homogeneous colorant solution was formed. The dyed fabric samples were prepared by soaking the cotton in the liquid dye for one hour at room temperature. The samples were then rinsed under cold water until the water ran clear. The dyed cotton was then dried completely before scanning. Cotton fabric swatches were received from The Museum of Fine Arts Houston.

Near-Infrared Raman Spectroscopy (NIRS): NIRS spectra were taken using a handheld Agilent Resolve spectrophotometer equipped with an 830 nm laser. The instrument had a beam diameter of roughly 2 mm. Spectra were acquired with a 1 s acquisition time and 495 mW laser power. The spectrophotometer performed an automatic baseline subtraction for every taken spectrum. In total, 25 spectra were taken for every dye and colored fabric. Raman spectra from the solution of dyes were taken in glass vials along with the vial holding feature employing a spatial offset modality of the Resolve instrument.

Spectral processing and Statistical analysis: Spectral processing was performed using Matlab (Mathworks, Natick, MA, USA) equipped with PLS_Toolbox (Eigenvector Research, Inc., Manson, WA, USA). All spectra were MSC-normalized and smoothed (SalGol). For PLS-DA, the model was used with a derivative (SalGov) (order: 2, window: 15 pt, tails: polyinterp) and mean centering. Kruskal–Wallis ANOVA was used to show significant differences between peak intensities. The number of latent variables (LVs) varied for each model. The number of LVs is reported for each PLS-DA models.

## 4. Conclusions

Our experimental results demonstrated that NIRS could be used for the non-invasive, non-destructive, and confirmatory identification of dyes on cotton fabric. Coupling PLS-DA with NIRS enabled ~100% accurate identification of 15 different colorants. We envision that the hand-held nature of the spectrometer used for this study will make it useful for the forensic analysis of fabric directly at a crime scene. We also anticipate that NIRS can be used by art conservation scientists to examine fabric in historical textiles and Gobelin rugs.

## Figures and Tables

**Figure 1 molecules-28-07864-f001:**
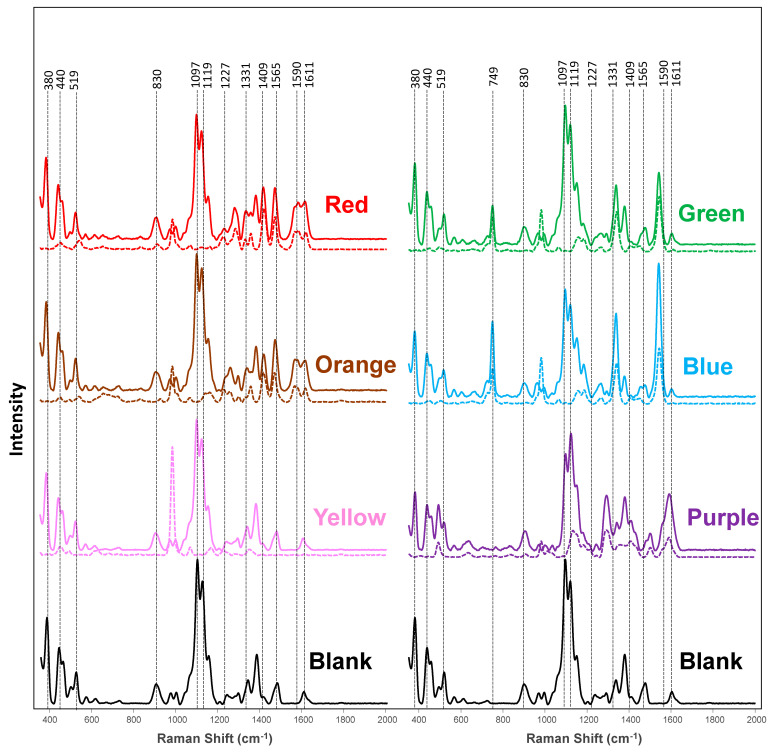
Averaged NIRS spectra acquired from the in-lab-dyed (solid lines) fabric and undyed cotton (blank) together with the corresponding spectra acquired from dyes dissolved in water (dashed lines).

**Figure 2 molecules-28-07864-f002:**
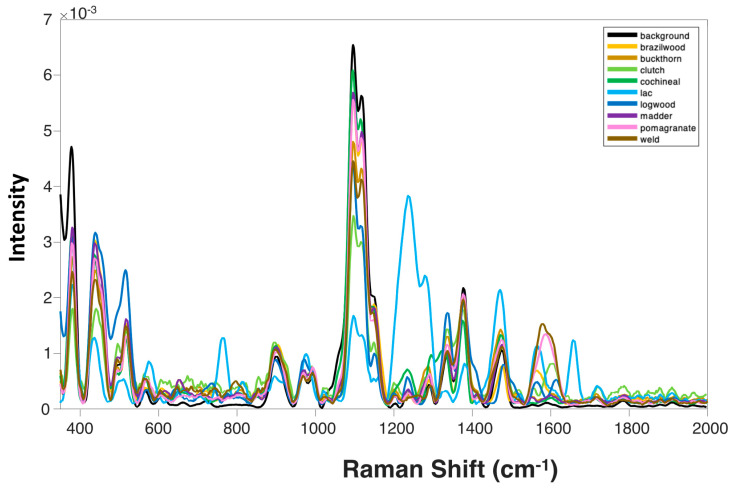
Averaged NIRS spectra of Menil fabric.

**Table 1 molecules-28-07864-t001:** Vibrational bands in the NIRS spectra acquired from in-lab-dyed and undyed cotton fabric.

Samples	Vibration Bands (cm^−1^)
Red	379, 437, 494, 519, 567, 611, 648 **, 720, 828 **, 902, 969, 995, 1095, 1119, 1150, 1227 **, 1278 **, 1331 **, 1353 **, 1377, 1413 **, 1469 **, 1579 *, 1611 **
Orange	380, 438, 454, 496, 520, 567, 611, 652 **, 723, 777 *, 829 **, 902, 969, 998, 1096, 1120, 1150, 1256 **, 1293, 1335, 1379, 1415 **, 1470, 1565 **, 1611 **
Yellow	380, 438, 455 *, 495, 519, 567, 610, 661 *, 724, 901, 969, 995, 1096, 1119, 1150, 1202 *, 1238, 1290, 1337, 1378, 1409 **, 1476, 1602
Green	379, 437, 454 *, 497, 518, 566, 606, 662 *, 724, 749 **, 815 *, 901, 966, 994, 1096, 1119, 1150, 1181 **, 1265 **, 1291, 1337, 1378, 1407 **, 1475, 1541 **, 1602
Blue	379, 437, 453 *,517, 566, 602 **, 662 *, 723, 749 **, 820 **, 900, 961 *, 993, 1032 *, 1095, 1119, 1183 **, 1263 **, 1292, 1337, 1377, 1406 **, 1455 *, 1474, 1540 **, 1601
Purple	380, 438, 455 *, 492, 518, 565, 701 **, 764 **, 833 **, 904, 969, 1000, 1033 **, 1097, 1123, 1148, 1243 **, 1291, 1341, 1379, 1407 **, 1500 **, 1590 **
Undyed	379, 437, 494, 518, 566, 610, 723, 900, 968, 995, 1095, 1118, 1149, 1237, 1289, 1336, 1378, 1475 and 1602

* = Evident in NIRS spectra of colored fabric but not in the NIRS spectra of blank cotton. ** = Evident in NIRS spectra of colored fabric and the spectra of dyes dissolved in water.

**Table 2 molecules-28-07864-t002:** PLS-DA results of identification of in-lab-dyed fabric (LV = 5).

	Actual Class	Accuracy	Red	Orange	Yellow	Green	Blue	Purple	White
Predicted Class	Red	100%	25	0	0	0	0	0	0
	Orange	100%	0	25	0	0	0	0	0
	Yellow	96%	0	0	24	0	0	0	1
	Green	100%	0	0	0	25	0	0	0
	Blue	100%	0	0	0	0	25	0	0
	Purple	100%	0	0	0	0	0	25	0
	Plain Cotton	84%	0	0	4	0	0	0	21

**Table 3 molecules-28-07864-t003:** PLS-DA results of differentiation between white and yellow in-lab-dyed fabric (LV = 2).

	Actual Class	Accuracy	Yellow	White
Predicted Class	Yellow	100%	25	0
	White	100%	0	25

**Table 4 molecules-28-07864-t004:** Vibrational bands in the NIRS spectra acquired from water-soluble colorants.

Dye	Vibration Bands (cm^−1^)
Red	447, 536, 649, 830, 908, 981, 1063, 1156, 1222, 1281, 1328, 1354, 1417, 1469, 1573 and 1613
Orange	379, 437, 454, 494, 519, 567, 611, 654, 722, 828, 901, 974, 995, 1096, 1119, 1150, 1236, 1255, 1292, 1336, 1378, 1414, 1470, 1565 and 1608
Yellow	447, 488, 616, 980, 1065, 1164, 1224, 1346, 1407 and 1609
Green	448, 496, 723, 749, 821, 981, 1064, 1158, 1183, 1265, 1340, 1406, 1448, 1543 and 1607
Blue	447, 501, 567, 605, 653, 727, 750, 821, 981, 1064, 1097, 1157, 1186, 1266, 1339, 1406, 1448, 1542 and 1604
Purple	406, 444, 491, 633, 702, 762, 833, 907, 981, 1011, 1132, 1181, 1209, 1244, 1291, 1356, 1407, 1500 and 1589

**Table 5 molecules-28-07864-t005:** PLS-DA results of identification of water-soluble colorants (LV = 5).

	Actual Class	Accuracy	Red	Orange	Yellow	Green	Blue	Purple
	Red	100%	25	0	0	0	0	0
Predicted Class	Orange	100%	0	25	0	0	0	0
	Yellow	100%	0	0	25	0	0	0
	Green	100%	0	0	0	25	0	0
	Blue	100%	0	0	0	0	25	0
	Purple	100%	0	0	0	0	0	25

**Table 6 molecules-28-07864-t006:** Vibrational bands in the NIRS spectra acquired from Menil fabric and the undyed cotton.

Samples	Vibration Bands (cm^−1^)
Brazilwood	380, 439, 519, 568, 610, 711 *, 848 *, 902, 970, 993, 1030 *, 1098, 1116, 1150 *, 1206 *, 1235, 1288, 1332, 1377, 1417, 1475, 1512 *, 1565 * and 1786
Buckthorn	381, 440, 519, 568, 609, 651 *, 739 *, 808 *, 853 *, 902, 970, 992, 1026 *, 1098, 1116, 1202 *, 1238, 1285, 1336, 1472, 1552 *, 1603 *, 1716 *, 1875 and 1914
Clutch	380, 442, 518, 567, 610, 677 *, 711 *, 748 *, 816 *, 858 *, 897, 969, 991, 1026 *, 1098, 1115, 1150 *, 1204 *, 1239, 1287, 1337, 1375, 1476, 1517 *, 1556 *, 1603 *, 1721 *, 1785, 1820 *, 1851 *, 1877 *, 1909 *, 1942 * and 1965 * cm^−1^
Cochineal	380, 439, 518, 567, 610, 663 *, 722, 760 *, 901, 968, 996, 1033 *, 1097, 1116, 1238, 1300 *, 1327 *, 1378, 1471, 1714 *, 1788 and 1913 * cm^−1^
Lac	381, 436, 504 *, 576 *, 659 *, 763 *, 813 *, 904, 977 *, 1102, 1160 *, 1237, 1275 *, 1382, 1471, 1567 *, 1658 * and 1721 * cm^−1^
Logwood	379, 439, 516, 569, 607, 644 *, 736 *, 777 *, 804 *, 851 *, 898, 971, 1033 *, 1096, 1116, 1151 *, 1200 *, 1236, 1292, 1337, 1379, 1481 *, 1564 *, 1613 *, 1714 *, 1780, 1880 *, 1915 * and 1964 * cm^−1^
Madder	380, 440, 518, 567, 616 *, 654 *, 803 *, 848 *, 901, 971, 992, 1098, 1116, 1236, 1287, 1335, 1378, 1414, 1475, 1601 * and 1787 cm^−1^
Pomegranate	380, 440, 518, 571, 607, 751 *, 806 *, 902, 971, 994, 1098, 1115, 1204 *, 1237, 1290, 1336, 1377, 1414, 1474, 1587 *, 1716 *, 1785 and 1914 cm^−1^
Weld	380, 440, 517, 565, 607, 650 *, 743 *, 798 *, 899, 971, 991, 1099, 1115, 1205 *, 1293, 1336, 1376, 1475 and 1580 * cm^−1^
Plain Cotton (Background)	378, 439, 518, 566, 609, 722, 901, 969, 994, 1097, 1116, 1238, 1288, 1336, 1378, 1413, 1474 and 1784 cm^−1^

* = Evident in NIRS spectra of colored fabric but not in the NIRS spectra of a blank fabric.

**Table 7 molecules-28-07864-t007:** PLS-DA results of identification of colored Menil fabric samples (LV = 7).

	Actual Class	Accuracy	Cotton (Undyed)	Eastern Brazilwood	Buckthorn	Clutch	Cochineal	Lac	Logwood	Madder	Pomegranate	Weld
Predicted Class	Cotton (Undyed)	92%	23	0	0	0	0	0	0	2	0	0
	Brazilwood	72%	0	18	0	0	0	0	0	7	0	0
	Buckthorn	84%	0	3	21	0	0	0	0	1	0	0
	Clutch	100%	0	0	0	25	0	0	0	0	0	0
	Cochineal	100%	0	0	0	0	25	0	0	0	0	0
	Lac	100%	0	0	0	0	0	25	0	0	0	0
	Logwood	100%	0	0	0	0	0	0	25	0	0	0
	Madder	84%	2	3	0	0	0	0	0	21	0	0
	Pomegranate	96%	0	0	0	0	0	0	0	0	24	1
	Weld	96%	0	0	0	0	0	0	0	0	1	24

## Data Availability

The data will be made available on reasonable request to the corresponding author.

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
