# Peer review of "Non-Destructive Identification of Dyes on Fabric Using Near-Infrared Raman Spectroscopy"

_molecules, 2023, doi:10.3390/molecules28237864_

Round 1

Reviewer 1 Report

Comments and Suggestions for Authors

 According to the structure of the journal, the material and methods section appear before the conclusion section. Please organise the article with this structure.

 In the discussion of results I do not advise to include bibliographical references that are not related to the results of the research. For example, information on AuNPs or dyes in oil- and crayon-paintings would be better if they were added in the introduction, considering that the article does not talk at all about these arguments.

Some references are repeated both in the introduction and in the discussion and do not provide new information. Put the bibliographical references in the introduction. If they are not useful to add information I recommend not repeat them.

Furthermore, I recommend doing a single paragraph of conclusions that also includes the discussion of the results.

Author Response

We want to thank the reviewer for the provided feedback and constrictive suggestions. 

1. In the discussion of results I do not advise to include bibliographical references that are not related to the results of the research. For example, information on AuNPs or dyes in oil- and crayon-paintings would be better if they were added in the introduction, considering that the article does not talk at all about these arguments.

Response: we removed references from the discussion.

2. Some references are repeated both in the introduction and in the discussion and do not provide new information. Put the bibliographical references in the introduction. If they are not useful to add information I recommend not repeat them.

Response: we moved all references from the discussion to the introduction.

3. Furthermore, I recommend doing a single paragraph of conclusions that also includes the discussion of the results.

Response: we added a conclusion paragraph to the manuscript.

Reviewer 2 Report

Comments and Suggestions for Authors

In the introduction, we don’t clearly link to the “case of William Wayne”, so we either suggest the authors to add a reference or to remove the name.

Page 2, line 64: the authors states that “our group previously…” but we don't understand what it refers to, given that the authors of the present text are different from those of the cited texts. We also find it superfluous to recall all the previous scientific production here.

Moreover, to validate the application of the method proposed by the authors in the forensic field, it seems appropriate to cite some published works such as Stoney et al. 2015 and 2016 on FSI, Merelli et al 2023 on IJLM and Cocks et al. 2015 on FSI.

In the material and methods, the process on the fabric material should be better explained: time, water temperature, the process followed by the manufacturing instruction.

Table 1 and table 7 should be made more usable for the reader.

Page 5, line 140-142: the solution that the authors find for the only non-optimal result of the study is explained in an excessively hasty way. Given that the results of 96% and 84% were already acceptable, the improvement of the method in order to reach 100% deserves to be further explored so as not to appear artificial.

The conclusions deserve to be expanded, both with the advantages from the scientific point of view of the method and for the possible developments and practical applications in the forensic field.

Author Response

In the introduction, we don’t clearly link to the “case of William Wayne”, so we either suggest the authors to add a reference or to remove the name.

Response: we removed the name.

Page 2, line 64: the authors states that “our group previously…” but we don't understand what it refers to, given that the authors of the present text are different from those of the cited texts. We also find it superfluous to recall all the previous scientific production here.

Response: we changed the sentence according to the provided suggestion.

Moreover, to validate the application of the method proposed by the authors in the forensic field, it seems appropriate to cite some published works such as Stoney et al. 2015 and 2016 on FSI, Merelli et al 2023 on IJLM and Cocks et al. 2015 on FSI.

Response: we want to thank the reviwer for the provided suggestion. We cited several research papers authored by Stoney and co-workers:

“Current methods for a fabric analysis primarily rely on a pattern recognition and are performed by an expert using a microscope (Houck and Siegel 2009; Stoney et al. 2016; Stoney et al. 2015; Stoney and Stoney 2015).”

In the material and methods, the process on the fabric material should be better explained: time, water temperature, the process followed by the manufacturing instruction.

Response: We expanded the description of the dying procedure.

Table 1 and table 7 should be made more usable for the reader.

Response: We changed the layout of these tables.

Page 5, line 140-142: the solution that the authors find for the only non-optimal result of the study is explained in an excessively hasty way. Given that the results of 96% and 84% were already acceptable, the improvement of the method in order to reach 100% deserves to be further explored so as not to appear artificial.

Response: To overcome this issue, we developed a separate PLS-DA model that was able to differentiate these two types of colored fabric with 100% accuracy, Table 3 and Figure S4.

The conclusions deserve to be expanded, both with the advantages from the scientific point of view of the method and for the possible developments and practical applications in the forensic field.

Response: we expanded the conclusion section of the manuscript.